# The Effect of Increasing Levels of Dehulled Faba Beans (*Vicia faba* L.) on Extrusion and Product Parameters for Dry Expanded Dog Food

**DOI:** 10.3390/foods8010026

**Published:** 2019-01-12

**Authors:** Isabella Corsato Alvarenga, Charles Gregory Aldrich

**Affiliations:** Grain Science & Industry, Kansas State University, Manhattan, KS 66506, USA; isacorsato@ksu.edu

**Keywords:** faba beans, dehulled, dog, pet food, extrusion

## Abstract

The growing pet food market is continuously in search for novel ingredients. Legumes such as faba beans are increasingly popular in human nutrition but have not yet been explored in pet foods. Extruded dog diets were produced with 0, 10, 20, and 30% dehulled faba bean (DFB) inclusion (FB0, FB10, FB20, and FB30, respectively) in exchange for rice and corn gluten meal. Fixed processing inputs were extruder screw configuration, die size (5.2 mm diameter), dry feed rate (237 kg/h), extruder water and steam (0%), and die knife speed (1100 rpm). Variable inputs were managed by an operator with the goal to obtain similar kibble bulk density at the extruder die (OE) across treatments. Output parameters were measured at the pre-conditioner, extruder, and kibble. Measurements were collected at uniform time increments during production of each experimental diet and considered treatment replicates. Single degree of freedom contrasts were analyzed on extrusion and product outputs. The target of producing diets with similar wet bulk density was achieved, with moderate modifications at the pre-conditioner (PC) and extruder. As DFB increased, diets had increased retention time and water at the PC to improve starch hydration and swelling. The FB20 and FB30 required a more restricted flow to improve kibble expansion. After drying, the FB20 and FB30 diets were denser, harder and tougher (*p* < 0.05) than FB0 and FB10. The increasing levels of DFB up to 30% can be effectively controlled in an extruded pet food application with modest changes to extrusion parameters.

## 1. Introduction

More than two-thirds of U.S. households have pets, with 60 and 47 million homes owning dogs and cats, respectively [1]. In total the pet food industry in the U.S. commands a significant market worth nearly $27 billion [2]. As this market matures, sales are mostly driven by novel ingredients. Many of these new ingredients have been introduced without published results with dog and cat feeding studies that evaluated their utilization, safety, or efficacy. Most of these ingredients are non-cereal starch sources such as peas, chickpeas, potatoes, and tapioca. These have been used in human foods for decades, if not centuries with little issue, as long as they are adequately processed. For instance, Alonso et al. [3] found that extrusion reduced condensed tannins by 91.7% and it also caused a large decrease in other antinutritional factors in peas (*Pisum sativa*). Hence, antinutritional factors in faba beans may not be a concern in pet extruded dry kibbles, but needs to be confirmed.

Legumes, like faba beans, are complimentary to cereals due to their high lysine content [4], and they are also rich in water soluble B vitamins [5]. Faba beans provide a wide range of benefits to human health due to phenolic compounds, oligosaccharides, enzyme inhibitors, phytosterols and saponins [6]. However, evidence is needed regarding how faba beans will process in extruded pet food applications, as a prelude to their evaluation for pet health and nutrition. Extrusion is the most common means to process pet foods. It comprises a cooking process that first conditions the dry recipe in a preconditioner, adding water, steam and mixing to the food, and then cooks and melts the dough in the extruder barrel under high shear, moisture, and pressure. At the end of the extruder barrel the pressure drops to atmospheric level as the dough exits the system, and kibble expansion occurs. Nutrients undergo physical changes during this cooking process, with starch gelatinization being the most relevant for kibble binding and structure formation. Different ingredients and their inclusion affect how the food is processed, so knowing the effect that increasing levels of dehulled faba beans have on process parameters is essential to produce homogeneous and aesthetically pleasing kibbles. Therefore, the objective of this work was to determine the effects of graded levels of dehulled faba beans (DFB) on extrusion processing parameters and final kibble consistency.

## 2. Materials and Methods

### 2.1. Diets

Dehulled ground faba beans, ground brewers rice, ground beet pulp pellets and corn gluten meal were all purchased from a local mill (Fairview Mills, LLC, Seneca, KS, USA). Four nutritionally complete experimental diets were formulated, in which rice served as the main carbohydrate source in the control. In the DFB diets, proportional quantities of rice and corn gluten meal were replaced at 10, 20, and 30% of the formula (Table 1). Diets were formulated to have similar protein, fat, vitamins and minerals content based on previous nutrient analyses. All ingredients were purchased in their ground form and mixed in a 136 kg capacity paddle mixer for 5 min. External markers chromic oxide (Cr_2_O_3_) and titanium dioxide (TiO_2_) were included at 0.25% and 0.4% levels in each experimental diet, for later use in a dog feeding study. The poultry fat and digest (dry dog flavor) were added topically to kibbles after drying.

### 2.2. Analytical Methods

The nutrient analyses of ration and post-extrusion dietary treatments were determined at a commercial analytical laboratory (Midwest Laboratories, Omaha, NE, USA) as composite samples for each treatment. These included: moisture and dry matter (AOAC 930.15), organic matter and ash (AOAC 942.05), crude protein (AOAC 990.03), fat by acid hydrolysis (AOAC 954.02, modified), crude fiber (AOCS Ba 6a-05), and minerals, including: calcium, phosphorus, potassium, magnesium, sodium, sulfur, copper, iron, manganese, and zinc (AOAC 985.01; modified). Short-chain oligosaccharides including sucrose, raffinose, stachyose and verbascose were measured according to an HPLC method described by Smiricky et al. [7], and biogenic amines were measured at a commercial laboratory (Midwest laboratories; Omaha, NE) according to procedures described by Shalaby [8], U.S.FDA [9], and El Aribi et al. [10].

### 2.3. Extrusion Processing

The dry expanded pet foods (Figure 1) were produced using a single screw extruder (model E525, ExtruTech, Inc., Sabetha, KS, USA). The preconditioner (model 16 × 72 DDC, ExtruTech, Inc., Sabetha, KS, USA) was configured with 12 beaters 45° back, 57 beaters in neutral position, in each of two shafts, and run at a speed of 160 rpm (Appendix A). The extruder had a defined profile (Appendix A) and barrel temperatures based on a typical commercial pet food configuration. The target in-barrel moisture was approximately 25% wet basis. Fixed input parameters were kept constant throughout all food production and included dry feed rate (237 kg/h), extruder (EX) water (0%), EX steam (0%) and knife speed (1100 rpm). Variable inputs were considered those controlled by the operator and included pre-conditioner (PC) speed, PC water, PC steam, EX screw speed and mass restriction valve (MRV).

Pre-conditioner and extruder parameters were all collected from sensor readouts during food production. Outputs were those parameters indirectly controlled by the input variables, and included PC discharge temperature, EX die temperature and pressure, total mass flow (TMF), wet bulk density, and specific mechanical energy (SME). Wet bulk density was measured off the extruder four times manually during each respective run, by filling a one-liter cup and leveling the kibbles with a metal ruler and weighing on a digital scale. Total mass flow was calculated by adding the total dry feed rate with water and steam injected in the preconditioner and extruder, assuming that 80% of the water coming from the EX steam is lost during flash-off, as kibbles exit the die:TMF = dry feed rate + PC water + 0.2 × PC steam + EX water + 0.2 × EX steam(1)

Specific mechanical energy (SME) was calculated according to the equation below [11]:
(2)$$\mathrm{SME}\;\left(\mathrm{kJ}|\mathrm{kg}\right)=\frac{\left(\tau -\tau o\right)/100\times \left(N|\mathrm{Nr}\right)\times \mathrm{Pr}}{m}$$
where τ is the % torque or motor load, τo is the no load torque, N is the screw speed, Νr is the rated screw speed (425 rpm), Ρr is the rated motor power (112 kW), and m is the total mass flow (kg/s). The ratio between N and Nr was corrected to be less than or equal to 1.

The size of the kibble was controlled through the extruder die opening (5.2 mm) in order to produce a kibble suitable for feeding small breed dogs. After extrusion, the product was dried on perforated trays in a forced air convection oven at approximately 141 °C until kibbles achieved a moisture level below 9%. The coating consisted of poultry fat and dry palatant and was applied to the kibbles in a double ribbon mixer after they were cooled.

Post extrusion, kibble radial expansion was measured using a digital caliper and expressed as sectional expansion index (SEI), which is the ratio of the cross-section area of the kibble to that of the extruder die. All measurements related to kibble characteristics were done by randomly selecting 10 kibbles per time point for each diet (4 replicates per treatment). Kibbles were weighed using a digital analytical balance, the diameter and length were measured using digital calipers, and these values were used to calculate piece density (g/cm^3^). Textural hardness and toughness of the products was measured using a texture analyzer (model TA-XT2, Texture Technologies Scarsdale, NY, USA) with a 2.54 cm-diameter cylinder probe. Subsamples of ten random kibbles per extrusion time point of each diet were compressed at 50% strain level. Pre-test speed was 2 mm/s, test speed 2 mm/s, and post-test speed was 10 mm/s. Kibble hardness was the highest breaking force necessary for compression at 50%, while toughness was considered the total force used to compress each kibble by 50%.

### 2.4. Statistical Analysis

The diets were produced in the order of FB0, FB10, FB20, and FB30 without randomization in an effort to maintain control over target bulk density. For each dietary treatment run, sampling was conducted at evenly spaced intervals and considered replicates for the purpose of determining variability and control during the test. Texture data were analyzed on the averages of the ten randomly selected kibbles from each replicate. Least square means of extrusion output responses were estimated by ANOVA using the GLM procedure with the aid of statistical software (SAS, v. 9.4, SAS Institute INC., Cary, NC, USA), using Tukey correction. Contrasts comparing “control vs. treatment”, linear, quadratic and cubic relationships of diets with graded levels of DFB were considered significant at a *p* < 0.05.

## 3. Results

### 3.1. Diets

The diet formulation premise was to maintain similar levels of protein, fat, minerals and vitamins across treatments, and this was met (Table 2). Vitamins were added at same concentration in all diets through the vitamin premix (Table 1). The diets were slightly drier than target (7–9%), but well within normal production parameters to avoid molding (Table 2). The crude protein content was higher than expected among all the diets and the level of fat was at or slightly lower than expected. Ash was below 6% for all diets, with minerals at similar levels across treatments.

Short-chain oligosaccharides were much higher in DFB than in rice, and as faba beans levels increased in the diets so did the oligosaccharides (Table 3). Within the tri- and tetrasaccharides, verbascose was the highest in faba bean (25,268 μg/g vs. 2620 μg/g in raffinose and 6967 μg/g stachyose). Biogenic amines histamine, putrescine and spermine were present in DFB before extrusion at low concentrations (12.1 ppm, 8.9 ppm and 17.4 ppm, respectively), and a similar amount of putrescine was present in all diets after extrusion (average 9.78 ppm; Table 4).

### 3.2. Extrusion Processing

The variable inputs during extrusion were manipulated by the operator to control the final product bulk density (Table 5). Pre-conditioner speed was higher when producing diets with lower faba bean inclusion (FB0 and FB10). Pre-conditioner water was slightly lower on the FB0 and FB10 compared to FB20 and FB30, but required more steam. Extruder screw speed was also increased in the FB20 and FB30, and the mass restriction valve (MRV) opening was decreased to increase barrel pressure. Fixed inputs were kept constant throughout all dietary production, as described before.

The goal of producing diets with similar wet bulk density was achieved (average 362.5 g/L; Table 6). Total mass flow while different among treatments was maintained in a very narrow range (271–275 kg/h). The SME was greater in the control than the other dietary treatments, and it decreased linearly (*p* < 0.05) as DFB inclusion increased. Pre-conditioner discharge temperature was lower in the control (FB0) compared to treatments, and increased linearly (*p* < 0.05) as DFB inclusion increased (Table 6). The extruder die temperature was lower (*p* < 0.05) in the control compared to treatments, and there was a linear and cubic (*p* < 0.05) increase at the higher DFB levels. Extruder die pressure was lower (*p* < 0.05) in the control treatment and increased linearly with greater DFB inclusion.

For the dried kibbles, piece density of the FB0 was not different from the other treatments, but kibbles became more dense (*p* < 0.05) as DFB inclusion increased (linear; *p* < 0.05; Table 6). This corresponded to the linear (*p* < 0.05) decline in SEI, linear increase (*p* < 0.05) in hardness, and quadratic increase (*p* < 0.05) in toughness as higher quantities of DFB were included in the diet.

## 4. Discussion

### 4.1. Diets

Dehulled faba bean crude protein and crude fat were close to 31% and 2%, respectively, which is similar to values previously reported in the literature [12,13,14]. Since faba beans are considered both a protein and starch ingredient, corn gluten meal and brewers rice were added in increasing concentrations in FB10, FB20 and FB30 to maintain similar nutrient levels across diets. Given the faba beans in the study were dehulled prior to receipt, the crude fiber was much lower than what has been reported in whole faba beans [15]. Dehulled faba beans in this study were used because condensed tannins are primarily concentrated in the seed hull or cotyledon surface. Some condensed tannins may under the right circumstances and in the right individuals cause favism, a blood disorder in humans [16]. These may be rendered inert when removed or destroyed during cooking processes [3].

Faba beans are known to contain measurable levels of other antinutritonal factors. Previous research reported that dehulling decreased the condensed tannins and polyphenol levels, and extrusion abolished trypsin, chymotrypsin, and α-amylase inhibitors and haemagglutinating activity [3]. However, the effect of extrusion processing on oligosaccharide sugars and biogenic amines seemed to be missing. Landry et al. [17] characterized the carbohydrate fractions of faba beans, and the mature whole seed had similar composition to the DFB of this study, with a higher concentration of sucrose and verbascose than raffinose and stachyose. However, the DFB had a higher amount of these sugars, likely because it did not contain the fibrous hull, which dilutes the short chain oligosaccharide concentrations. Biogenic amines are antinutritional factors that initiate several pharmacological reactions which promote a number of food poisoning episodes in humans and animals [8]. The most important biogenic amines present in legumes are histamine, putrescine, cadaverine, tyramine, tryptamine, 2-phenylethylamine, spermine, and spermidine. A study found that in faba beans all these biogenic amines were below detection limits after cooking [18]. In our study only histamine, putrescine and spermine were detected in DFB and their amounts were low enough that they could be considered insignificant.

### 4.2. Extrusion Processing

This study was a prelude to evaluation of faba bean diets being fed to dogs. Capturing this processing information was vital to gaining a full understanding of how this ingredient affects their use in pet foods. In this work, the DFB extruded diets were produced successfully to target finished product characteristics. Namely, a similar wet bulk density and sufficient expansion to produce a food consistent with commercial products and acceptable for feeding to dogs in the study that followed. The limitation of the extrusion processing was that diets were produced in a single production sequence with multiple samples over time; however, it would be ideal in the future to evaluate faba bean containing diets in multiple start-stop cycles and on different extruder models. All diets met or exceeded the target nutritional levels of protein, fat, vitamins and minerals (Table 1 and Table 2). Extruded kibbles with graded levels of DFB had a consistent, stable production, and diets achieved similar bulk density, with minor to moderate modifications in the pre-conditioner and extruder settings. An inclusion of 10% DFB did not have an effect on processing or final product characteristics, with similar pre-conditioner and extruder inputs to the control. Faba bean produces a stickier flour [19] which is solubilized at lower temperatures [19,20], and resists swelling more when compared to cereals [19]. Therefore, in the present study some pre-conditioner (PC) and extruder (EX) variables had to be changed by the operator as DFB levels increased in order to compensate for their different physical attributes. By lowering the PC speed the retention time was increased, and that combined with higher PC water addition contributed to a greater starch cooking and swelling. Additionally, the decrease in the MRV opening when extruding the FB20 and FB30 diets improved expansion off the extruder, ostensibly due to the addition of legumes to the recipe that tend to lower expansion when used as replacement of cereals [21]. All the processing modifications to increase retention time and cook of diets containing more faba beans (FB20 and FB30) led to an increase in the EX die pressure and temperature (Table 6).

The output variables measured at the PC and EX were indirectly altered by the fixed and input variables. As an outcome of increasing PC water, the PC temperature increased in a linear fashion. The increase in EX die temperature when producing the FB20 and FB30 diets also reflects the higher EX screw speed set when producing these diets. The increase in the EX screw speed did not correspond to an increase in SME of these diets compared to the control.

Kibbles with higher DFB inclusion were denser and less expanded after drying, which resulted in harder and tougher kibbles. Anton et al. [22] also found that corn extrudates were more dense and less expanded when navy and small red beans inclusion increased in the recipe. Similarly, Patil et al. [23] reported an increase in ground lentils extrudate sectional expansion index (SEI) when starch was added. The reason why pulses like faba beans produce denser and less expanded extrudates is still unknown, and it may be due to their starch granule structure and crystallinity. In faba beans the starch granules are “cracked” with a C type x-ray pattern, and they also have a lower pasting (amylose leaching) temperature [24]. In the present study toughness had a large variation, but there was still a significant quadratic relationship among treatments. Alvarenga et al. [25] also found a high variation in toughness of extruded dog kibbles. This might be due to the nature of the expanded kibble, which has non-uniform air cells in its interior and therefore each kibble is compressed differently. Also, more replication would be needed to reduce variability and increase precision of this analysis. Piece hardness also had a quadratic relationship among diets, but there was no difference between the control vs treatments. Although kibbles had similar wet bulk densities, the drying effect on kibbles with a higher DFB inclusion (FB20 and FB30) led to harder, tougher and denser kibbles.

## 5. Conclusions

The present study showed that it is possible to include up to 30% DFB in the diet and produce dog kibbles with similar wet bulk density to the control. This requires moderate modifications in the process, and flow restriction at the extruder die seems to be the greatest contributor to expansion. Kibble shrinking after drying was observed, and this led to harder and tougher kibbles as DFB increased. Process variables may be controllable, but DFB inclusions over 10% may require additional evaluation in order to validate product characteristics for pet and owner confidence in this new ingredient.

## Figures and Tables

**Figure 1 foods-08-00026-f001:**
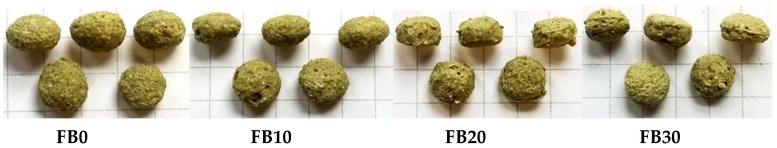
Dog kibbles produced containing graded levels of dehulled faba beans.

**Table 1 foods-08-00026-t001:** Ingredient composition of dietary treatments containing incremental levels of dehulled faba beans (FB0, FB10, FB20, and FB30, respectively) in a high protein dog diet.

Ingredient, %	FB0	FB10	FB20	FB30
Faba Beans, Dehulled	0.00	10.00	20.00	30.00
Rice, Brewers	44.59	37.90	32.00	26.10
Chicken Meal, Low Ash	31.85	28.98	28.91	28.84
Corn Gluten Meal, 60%	10.00	9.14	5.20	1.25
Beet Pulp	4.00	4.00	4.00	4.00
Salt	0.650	0.650	0.650	0.650
Potassium Chloride	0.325	0.250	0.250	0.250
Choline Chloride, 60% dry	0.200	0.200	0.200	0.200
Dicalcium Phosphate	0.033	0.171	0.108	0.045
Fish Oil	0.145	0.145	0.144	0.144
Dry Natural Antioxidant	0.034	0.033	0.033	0.033
Liquid Natural Antioxidant	0.031	0.033	0.033	0.033
Vitamin Premix	0.150	0.150	0.150	0.150
Trace Mineral Premix	0.100	0.100	0.100	0.100
Chromium Sesquioxide	0.250	0.250	0.250	0.250
Titanium Dioxide	0.400	0.400	0.400	0.400
Chicken Fat (topical)	6.25	6.61	6.58	6.56
Digest, Dry Dog Flavor (topical)	1.00	1.00	1.00	1.00

**Table 2 foods-08-00026-t002:** Nutrient analysis on a dry matter basis of raw dehulled faba beans and post-extrusion experimental diets with 0%, 10%, 20% and 30% dehulled faba beans (FB0, FB10, FB20 and FB30, respectively).

Item	Dehulled Faba Beans	FB0	FB10	FB20	FB30
Dry Matter, %	89.34	95.69	94.08	94.28	96.5
Crude Protein, %	30.8	36.3	36.6	37.5	38.1
Crude Fat, %	1.75	13.4	12.5	14.2	12.1
Crude Fiber, %	0.42	2.06	3.61	1.16	3.23
Ash, %	3.30	5.54	5.53	5.60	6.02
Calcium, %	0.12	0.91	0.91	0.91	0.88
Phosphorous, %	0.58	0.76	0.80	0.73	0.77
Potassium, %	1.34	0.63	0.74	0.59	0.80
Magnesium, %	0.15	0.09	0.10	0.08	0.1
Sodium, %	0.01	0.47	0.47	0.48	0.44
Sulfur, %	0.24	0.44	0.42	0.46	0.37
Manganese, ppm	21.7	21.3	21.9	21.1	18.3
Copper, ppm	15.6	18.2	18.9	17.7	19.1
Iron, ppm	76.7	124	128	127	117
Zinc, ppm	61.9	140	135	130	138

**Table 3 foods-08-00026-t003:** Short-chain oligosaccharides measured in brewers rice, dehulled faba bean, and experimental diets with 0%, 10%, 20% and 30% dehulled faba beans (FB0, FB10, FB20 and FB30, respectively).

Item, μg/g	Brewers Rice	Dehulled Faba Beans	FB0	FB10	FB20	FB30
Sucrose	1711	33,468	3098	5744	9088	10,920
Raffinose	81	2620	108	321	547	787
Stachyose	0	6967	23	673	1423	2037
Verbascose	0	25,268	0	2459	4968	7430

**Table 4 foods-08-00026-t004:** Biogenic amines (as is basis) in dehulled faba beans and diets FB0, FB10, FB20 and FB30 post-extrusion.

Item, ppm	Dehulled Faba Beans	FB0	FB10	FB20	FB30
2-phenylethylamine	n.d.	n.d.	n.d.	n.d.	n.d.
Cadaverine	n.d.	n.d.	n.d.	n.d.	n.d.
Histamine	12.1	n.d.	n.d.	n.d.	n.d.
Putrescine	8.9	9.1	9.2	9.5	11.3
Spermidine	n.d.	5.5	n.d.	5.6	n.d.
Spermine	17.4	n.d.	n.d.	n.d.	n.d.
Tryptamine	n.d.	n.d.	n.d.	n.d.	n.d.
Tyramine	n.d.	n.d.	n.d.	n.d.	n.d.

n.d. = not detected; ppm = parts per million; ppm = mg/kg.

**Table 5 foods-08-00026-t005:** Variable inputs of the pre-conditioner (PC) and extruder (EX) used to produce dehulled faba beans diets at 0, 10, 20, and 30% inclusion (FB0, FB10, FB20 and FB30), reported as average ± standard deviation.

Item	FB0	FB10	FB20	FB30
^1^N	8	6	7	4
^2^PC speed, rpm	185 ± 0	182 ± 8.16	165 ± 0	165 ± 0
^2^PC water, kg/h	20.7 ± 0.49	21.3 ± 1.79	24.7 ± 0.24	24.6 ± 0.13
^2^PC steam, kg/h	69.0 ± 4.09	82.0 ± 10.72	60.8 ± 0.72	61.8 ± 0.80
^3^EX screw speed, rpm	481 ± 34.7	500 ± 0	521 ± 9.45	525 ± 0
^4^MRV, %	55.0 ± 9.26	50.0 ± 0	37.1 ± 5.67	35.0 ± 0

^1^N = number of replicates (sampling times during one run) for each treatment; ^2^PC = pre-conditioner; ^3^EX = extruder; ^4^MRV = Mass restriction valve.

**Table 6 foods-08-00026-t006:** Least square means and contrasts (FB0 vs FB10-30(T), linear (L); quadratic (Q); cubic (C) level of FB) for outputs from the processing parameters used to produce dehulled faba beans (FB) diets at 0, 10, 20, and 30% inclusion (FB0, FB10, FB20 and FB30).

Item	FB0	FB10	FB20	FB30	MSE	*FB0* vs. *T*	*L*	*Q*	*C*
^1^N	8	6	7	4					
Wet Bulk density, g/L	365	358	364	363	49.44	0.2571	0.9348	0.2780	0.1369
^2^TMF, kg/h	271	275	274	274	0.1	<0.0001	<0.0001	<0.0001	<0.0001
^3^SME, kJ/kg	187	157	157	145	593	0.0042	0.0133	0.4093	0.3336
^4^PC discharge temp, °C	66.2	70.2	72.5	75.0	2.89	<0.0001	<0.0001	0.3062	0.5827
^5^EX die temp, °C	104	110	146	144	48.1	<0.0001	<0.0001	0.2007	<0.0001
^5^EX die pressure, MPa	3.16	3.28	3.29	3.46	0.016	0.0027	0.0011	0.6668	0.2440
^1^N	4	3	4	3					
Piece density, g/cm^3^	0.426	0.398	0.476	0.517	0.0009	0.0574	0.0006	0.0525	0.0679
^6^Piece SEI (dry), mm^2^/mm^2^	2.82	2.90	2.67	2.52	0.017	0.1552	0.0059	0.1398	0.2570
Piece hardness, kg	8.15	6.93	8.35	10.20	1.521	0.6500	0.0295	0.0440	0.4780
Piece toughness, kg·mm	767	360	601	678	31,141.3	0.0617	0.9537	0.0296	0.0860

^1^N = number of replicates (sampling times during one run) for each treatment; ^2^TMF = total mass flow; ^3^SME = specific mechanical energy; ^4^PC = pre-conditioner; ^5^EX = extruder; ^6^SEI = sectional expansion index.

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
