# Peer review of "The Effect of Increasing Levels of Dehulled Faba Beans (Vicia faba L.) on Extrusion and Product Parameters for Dry Expanded Dog Food"

_foods, 2019, doi:10.3390/foods8010026_

Round 1
Reviewer 1 Report
1// In introduction, state of art need to report in detail way.
2// Subheadings are required in materials and methods section. As example, line 65, analytical methods. Examples are not limited.
3// Table 2 and Table 3 are not necessary to address in material and method section.
4// Allergy is a major concern for pet food. Authors need to include the allergenic activity of end product. Also, in results and discussion section, discussion about it is demanding.
5// Results of biogenic amine in table is necessary
Author Response
Open Review
(x) I would not like to sign my review report
( ) I would like to sign my review report
English language and style
( ) Extensive editing of English language and style required
( ) Moderate English changes required
(x) English language and style are fine/minor spell check required
( ) I don't feel qualified to judge about the English language and style
Yes | Can be improved | Must be improved | Not applicable | |
Does the introduction provide sufficient background and include all relevant references? | ( ) | ( ) | (x) | ( ) |
Is the research design appropriate? | ( ) | (x) | ( ) | ( ) |
Are the methods adequately described? | ( ) | ( ) | (x) | ( ) |
Are the results clearly presented? | ( ) | ( ) | (x) | ( ) |
Are the conclusions supported by the results? | (x) | ( ) | ( ) | ( ) |
Comments and Suggestions for Authors
1// In introduction, state of art need to report in detail way. The authors are very close to the new product trends in the pet food space. Consideration for use of faba beans is state of the art. A subtle change to sentences in L34 and L35 were made to help accentuate this.
2// Subheadings are required in materials and methods section. As example, line 65, analytical methods. Examples are not limited. Thank you for your comment. We have added a subheading “2.2 Analytical methods”.
3// Table 2 and Table 3 are not necessary to address in material and method section. We agree that the Tables are more appropriate under “Results”. Therefore, we have moved Tables 2 to 5 to the Results section.
4// Allergy is a major concern for pet food. Authors need to include the allergenic activity of end product. Also, in results and discussion section, discussion about it is demanding.The goal of this processing manuscript was solely to focus on the processing of faba beans in extruded kibbles. A feeding study was conducted following diet production and this data will be presented in the manuscript reporting this work. Perhaps this area of allergens/sensitizing elements would be better addressed in that context.
5// Results of biogenic amine in table is necessary.Thank you for your comment. Initially we removed the biogenic amines table because we wanted to condense the number of tables. It has been returned.

Reviewer 2 Report
Foods-399207
Overall comments
· Title of the manuscript does not clearly represent the focus of the study.
· The authors frequently addressed in this manuscript that some parameters (Table 4 - pre-conditioner (PC) speed, PC water, PC steam, EX screw speed and mass restriction valve) were controlled by the operator when dealing with different diet formulations. This is a very subjective decision, I am doubtful if these parameters are reproducible with different extrusion runs when handling the same formulation. The consistency of the final product characteristics needs to be validated with different runs under these variable input parameters. In particular, the authors have clearly stated that some output parameters (Table 5 - PC discharge temperature, EX die temperature and pressure, total mass flow (TMF), wet bulk density, and specific mechanical energy) can be indirectly controlled by the input variables. It will be interesting to check if it is possible to produce kibbles with increasing levels FB without manipulating these inputs and evaluate how they differ in the final product quality characteristics. This will help to explain some of the phenomena observed in the product whether it is attributed by the FB levels or the different extrusion conditions manipulated for each formulations. Starch and protein can behave very differently under different extrusion conditions. The thermal stability of the different FB formulations is also worth investigating – this will provide some information on how the food components respond under temperature influence.
· The level of common antinutrients in faba beans (e.g. trypsin inhibitors, condensed tannins, phytic acid, saponins, lectins, and favism-inducing factors (vicine and convicine)) in each of experimental diet should be evaluated or at least commented in the discussion section.
· Please fix the references - some references are not numbered (e.g. Line 116)
Specific comments
Please add more details about extrusion as a processing technique in making pet food in Introduction section.
Page 2 – Lines 56-57: Please specify the manufacturer and model for capacity paddle mixer.
Table 1 – Is there any reason the faba bean was not formulated beyond 30%?
Lines 55-56, 159-164 – Please support these statements. E.g. perform a statistical analysis on the nutrient analysis (Table 2) comparing the 4 dietary treatments.
Lines 57-59, 218 – Please cite the dog feeding study.
Page 3 – Lines 70-71: Please mention short-chain oligosaccharides are measured using what analytical technique.
Tables 2, 3 and 4 – Please report the statistical analysis (e.g. F and p values from ANOVA test) comparing the value for each experimental diets.
Tables 4 and 5 – Please define the row “N”. Is this the number of treatment replicates or number of samples collected within the same run? Why N ranged from 3 and 8.
Table 5 – Please define the column FB0 vs T, L, Q and C. Also, please comment the EX die temperature (FB20 and FB30>FB0 and FB10) in the discussion.
Page 6 – Lines 133-134: Please specify the drying duration and the final moisture content achieved
Page 6 – Lines 142-146: Please clarify how to measure/calculate product toughness.
Lines 161, 220-221: Not clear, please specific what is your “target” value or product characteristics.
Page 6 – Lines 168-170: Please present the biogenic amine result, either as separate result table in the manuscript or as supplementary material.
Page 6 – Lines 173-178: Please support this statement by reporting the statistical analysis in Table 4.
Page 7 – Lines 213-25: Please support this statement with a reference.
Page 7 – Lines223-225: Please support this statement with a reference. Please specify the time intervals (in xx seconds, minutes or hours??) you have collected the samples.
With respect to the toughness of the kibbles, please comment the fact that FB20 and FB0 has the same hardness, FB10 was less hard than FB0.
Author Response
Open Review
(x) I would not like to sign my review report
( ) I would like to sign my review report
English language and style
( ) Extensive editing of English language and style required
( ) Moderate English changes required
(x) English language and style are fine/minor spell check required
( ) I don't feel qualified to judge about the English language and style
Yes | Can be improved | Must be improved | Not applicable | |
Does the introduction provide sufficient background and include all relevant references? | ( ) | (x) | ( ) | ( ) |
Is the research design appropriate? | ( ) | (x) | ( ) | ( ) |
Are the methods adequately described? | (x) | ( ) | ( ) | ( ) |
Are the results clearly presented? | ( ) | (x) | ( ) | ( ) |
Are the conclusions supported by the results? | (x) | ( ) | ( ) | ( ) |
Comments and Suggestions for Authors
Foods-399207
Overall comments
· Title of the manuscript does not clearly represent the focus of the study.We have reworded the title to the following in an attempt to address this concern and better describe the intent for the paper: The Effect of Increasing Levels of Dehulled Faba Beans (Vicia faba L.) on Extrusion and Product Parameters for Dry Expanded Dog Food. Please advise if this misses the mark. Thank you.
· The authors frequently addressed in this manuscript that some parameters (Table 4 - pre-conditioner (PC) speed, PC water, PC steam, EX screw speed and mass restriction valve) were controlled by the operator when dealing with different diet formulations. This is a very subjective decision, I am doubtful if these parameters are reproducible with different extrusion runs when handling the same formulation. The consistency of the final product characteristics needs to be validated with different runs under these variable input parameters. In particular, the authors have clearly stated that some output parameters (Table 5 - PC discharge temperature, EX die temperature and pressure, total mass flow (TMF), wet bulk density, and specific mechanical energy) can be indirectly controlled by the input variables. It will be interesting to check if it is possible to produce kibbles with increasing levels FB without manipulating these inputs and evaluate how they differ in the final product quality characteristics. This will help to explain some of the phenomena observed in the product whether it is attributed by the FB levels or the different extrusion conditions manipulated for each formulations. Starch and protein can behave very differently under different extrusion conditions. The thermal stability of the different FB formulations is also worth investigating – this will provide some information on how the food components respond under temperature influence.
We agree it would be ideal to run the treatments multiple times and in more than one extruder, but unfortunately this was a limitation of the study. We have attempted to state this in the discussion. These output parameters are known to be manipulated by other inputs. For example, temperature will increase if we increase steam, and total mass flow will increase if we increase dry feed rate, and so on. Also, since faba beans affect the viscosity of the dough, and our goal was to produce kibbles with similar bulk density in all the treatments, it was necessary for the operator to make some adjustments as we changed recipes. The product as noted in the materials and methods were produced on a “production” sized machine, and most of the parameters were noted in the methods, along with the output of each. Thus, we feel fairly confident that the results could be reproduced. However, this was not a premise for the original work and would be something that if funded we would very much like to confirm.
· The level of common antinutrients in faba beans (e.g. trypsin inhibitors, condensed tannins, phytic acid, saponins, lectins, and favism-inducing factors (vicine and convicine)) in each of experimental diet should be evaluated or at least commented in the discussion section. The work of Alonso et al., 2000 had previously reported on the anti-nutritional properties. However, they did not describe the oligosaccharides or the biogenic amines. We had a question regarding their content and how the process would influence the same. Thus, these were measured in our project. We will address this in the discussion.
· Please fix the references - some references are not numbered (e.g. Line 116).Thank you for noticing. This has been fixed.
Specific comments
Please add more details about extrusion as a processing technique in making pet food in Introduction section. Thank you for your suggestion. We have included in the introduction: “It comprises a cooking process that first conditions the dry recipe in a preconditioner, adding water, steam and mixing to the food, and then cooks and melts the dough in the extruder barrel under high shear, moisture, and pressure. At the end of the extruder barrel the pressure drops to atmospheric and kibble expansion occurs. Nutrients undergo physical changes during this cooking process, with starch gelatinization being the most relevant for kibble binding and structuring. Different ingredients and their inclusion affect how the food is processed, so knowing the effect that increasing levels of dehulled faba beans have on process parameters is essential to produce homogeneous and aesthetically pleasing kibbles.”
Page 2 – Lines 56-57: Please specify the manufacturer and model for capacity paddle mixer. This paddle mixer was customized for the extruder operator, so this information is not available to us.
Table 1 – Is there any reason the faba bean was not formulated beyond 30%? Yes. Pet food manufacturers usually include up to 15% legume seeds in their formulation and 30% is considered an extreme. Exceeding this would begin to move into the realm of human snack food production which was outside the scope of this project.
Lines 55-56, 159-164 – Please support these statements. E.g. perform a statistical analysis on the nutrient analysis (Table 2) comparing the 4 dietary treatments.
L61-62 was revised to: “Diets were formulated to have similar protein, fat, vitamins and minerals contents based on previous nutrient analyses.”
The proximate composition for the ration and the finished kibble were composites of sub-samples taken at intervals during production.
On L144-145 we have modified: “The diet formulation premise was to maintain similar levels of protein, fat, minerals and vitamins across treatments, and this was met (Table 2). Vitamins were added at same concentration in all diets through the vitamin premix (Table 1).”
Lines 57-59, 218 – Please cite the dog feeding study.We have not submitted this paper for publication as of this date. When that study is published we will most assuredly reference this paper to close the loop.
Page 3 – Lines 70-71: Please mention short-chain oligosaccharides are measured using what analytical technique. We have modified: Short-chain oligosaccharides sucrose, raffinose, stachyose and verbascose were measured according to an HPLC method described by Smiricky et al. [7].
Tables 2, 3 and 4 – Please report the statistical analysis (e.g. F and p values from ANOVA test) comparing the value for each experimental diets. The faba beans were from a single source or lot and were included into each treatment at incremental levels. Thus, the analyses provided are intended to characterize the composition of the diets as produced. To perform statistics on replicates from each single batch of ration would merely be an artificial construct. To address this in future work it might be worthwhile to secure multiple batches of faba beans, produce them in multiple batches of ration, and produce the diets on the extruder over multiple start-stop cycles. However, this was outside the scope of what this project was intended to evaluate. We appreciate the desire for replication and statistics on this area. But, feel this approach at reporting the information is consistent with the objectives of the experiment which was focused on the inputs and outputs during processing.
Tables 4 and 5 (tables 5 & 6 now)– Please define the row “N”. Is this the number of treatment replicates or number of samples collected within the same run? Why N ranged from 3 and 8.“N” is the number of replicates, which in this study was considered to be samples collected in regular time points during each diet production. Some diets had a longer production time, so they had more collection points. We have added below Tables 5 & 6: 1N= number of replicates (sampling times during one run) for each treatment.
Table 5 – Please define the column FB0 vs T, L, Q and C. We have defined it on the Table title: “Table 6.Least square means and contrasts (FB0 vs T, FB0 vs treatment; L, linear; Q, quadratic; C, cubic) of outputs of the processing parameters used to produce dehulled faba beans diets at 0, 10, 20, and 30% inclusion (FB0, FB10, FB20 and FB30).” Also, please comment the EX die temperature (FB20 and FB30>FB0 and FB10) in the discussion. That was a great suggestion. We have added on L257-259: “All the processing modifications to increase retention time and cook of diets containing more faba beans (FB20 and FB30) led to an increase in the EX die pressure and temperature (Table 6).”
Page 6 – Lines 133-134: Please specify the drying duration and the final moisture content achieved. Diets were dried for the time necessary to obtain a moisture less than 9%. We have modified L 117-118: “After extrusion, the product was dried on perforated trays in a forced air convection oven at approximately 141 °C until kibbles achieved a moisture level below 9%.”
Page 6 – Lines 142-146: Please clarify how to measure/calculate product toughness. Thank you for noting this detail. We added on L130-131: Kibble hardness was the highest breaking force necessary for compression at 50%, while toughness was considered the total force used to compress each kibble by 50%.
Lines 161, 220-221: Not clear, please specific what is your “target” value or product characteristics. On L. 146 we added “drier than target (7-9%)”.This has been addressed in the manuscript (L220-221) and discussed above.
Page 6 – Lines 168-170: Please present the biogenic amine result, either as separate result table in the manuscript or as supplementary material. We have included this data in a Table (Table 4).
Page 6 – Lines 173-178: Please support this statement by reporting the statistical analysis in Table 4. As discussed above, statistical analysis was not performed on this data.
Page 7 – Lines 213-25: Please support this statement with a reference. This has been modified as part of a prior comment. The sentence in question has been removed.
Page 7 – Lines223-225: Please support this statement with a reference. Please specify the time intervals (in xx seconds, minutes or hours??) you have collected the samples. Samples were collected every 20 minutes. This paragraph has been reworded to better describe the process.
With respect to the toughness of the kibbles, please comment the fact that FB20 and FB0 has the same hardness, FB10 was less hard than FB0. We included on L277-278: Piece hardness also had a quadratic relationship among diets, but there was no difference between the control vs treatments. Although kibbles had similar wet bulk densities, the drying effect on kibbles with a higher DFB inclusion (FB20 and FB30) led to harder, tougher and denser kibbles.

Round 2
Reviewer 1 Report
I am satisfied with the justifications provided by authors. Corrections in the manuscript are appreciable.
Author Response
Open Review
(x) I would not like to sign my review report
( ) I would like to sign my review report
English language and style
( ) Extensive editing of English language and style required
( ) Moderate English changes required
(x) English language and style are fine/minor spell check required
( ) I don't feel qualified to judge about the English language and style
Yes | Can be improved | Must be improved | Not applicable | |
Does the introduction provide sufficient background and include all relevant references? | (x) | ( ) | ( ) | ( ) |
Is the research design appropriate? | (x) | ( ) | ( ) | ( ) |
Are the methods adequately described? | (x) | ( ) | ( ) | ( ) |
Are the results clearly presented? | (x) | ( ) | ( ) | ( ) |
Are the conclusions supported by the results? | (x) | ( ) | ( ) | ( ) |
Comments and Suggestions for Authors
I am satisfied with the justifications provided by authors. Corrections in the manuscript are appreciable.
Thank you. We have reread the manuscript and made it more clear.

Reviewer 2 Report
Thank you for addressing all the comments raised. Please change the title of this manuscript to "The Effect of Increasing Levels of Dehulled Faba Beans (Vicia faba L.) on Extrusion and Product Parameters for Dry Expanded Dog Food"
One additional comment for Table 2 - Lines 184-185: The crude protein content was higher than expected among all the diets and the level of fat was at or slightly lower than expected. Please check the accuracy of this sentence as the crude fat% for dehulled faba bean is very low as reported in Table 2.
Author Response
Open Review
(x) I would not like to sign my review report
( ) I would like to sign my review report
English language and style
( ) Extensive editing of English language and style required
( ) Moderate English changes required
(x) English language and style are fine/minor spell check required
( ) I don't feel qualified to judge about the English language and style
Yes | Can be improved | Must be improved | Not applicable | |
Does the introduction provide sufficient background and include all relevant references? | (x) | ( ) | ( ) | ( ) |
Is the research design appropriate? | (x) | ( ) | ( ) | ( ) |
Are the methods adequately described? | (x) | ( ) | ( ) | ( ) |
Are the results clearly presented? | (x) | ( ) | ( ) | ( ) |
Are the conclusions supported by the results? | (x) | ( ) | ( ) | ( ) |
Comments and Suggestions for Authors
Thank you for addressing all the comments raised. Please change the title of this manuscript to "The Effect of Increasing Levels of Dehulled Faba Beans (Vicia faba L.) on Extrusion and Product Parameters for Dry Expanded Dog Food"
That was a good suggestion. We have changed the title for what was suggested above.
One additional comment for Table 2 - Lines 184-185: The crude protein content was higher than expected among all the diets and the level of fat was at or slightly lower than expected. Please check the accuracy of this sentence as the crude fat% for dehulled faba bean is very low as reported in Table 2.
While it is true that the faba bean fat levels are low, the majority of the fat would be coming from the topical application of chicken fat and not from the beans. We attempted to maintain consistent protein, fat, starch levels among these diets. The crude protein content of the faba beans and the other constituents of the diet were generally higher than what had been used to predict the outcome of the formula based on book values and some assumptions.
